# A Novel Hybrid Ant Colony Optimization for a Multicast Routing Problem

**Xiaoxia Zhang \*, Xin Shen and Ziqiao Yu**

College of Software Engineering, University of Science and Technology LiaoNing, Anshan 114051, China;
yxy@ustl.edu.cn (X.S.); orfilaperpetual@gmail.com (Z.Y.)

**\*** Correspondence: zhangxiaoxia@ustl.edu.cn; Tel.: +86-188-4121-9708

**Abstract:** Quality of service multicast routing is an important research topic in networks. Research has sought to obtain a multicast routing tree at the lowest cost that satisfies bandwidth, delay and delay jitter constraints. Due to its non-deterministic polynomial complete problem, many meta-heuristic algorithms have been adopted to solve this kind of problem. The paper presents a new hybrid algorithm, namely ACO&CM, to solve the problem. The primary innovative point is to combine the solution generation process of ant colony optimization (ACO) algorithm with the Cloud model (CM). Moreover, within the framework structure of the ACO, we embed the cloud model in the ACO algorithm to enhance the performance of the ACO algorithm by adjusting the pheromone trail on the edges. Although a high pheromone trail intensity on some edges may trap into local optimum, the pheromone updating strategy based on the CM is used to search for high-quality areas. In order to avoid the possibility of loop formation, we devise a memory detection search (MDS) strategy, and integrate it into the path construction process. Finally, computational results demonstrate that the hybrid algorithm has advantages of an efficient and excellent performance for the solution quality.

**Keywords:** ant colony optimization; multicast routing; memory detection search; cloud model

## 1. Introduction

At present, with the quick development of networking applications, and the exponentially growing requirement for high speed data transmission in communication networks, realizing network-routing in the field of network and distributed systems has gradually become an important research topic. The implementation of network-routing can usually be divided into three modes: unicast mode, broadcast, and multicast mode. Implementation of the unicast mode sends the information data from a data source node to a demanding destination. The demand occurs if the source is asked privately by the end-user. This is the most common, one-to-one transmission of unicast on the Internet. On the other hand, implementations of broadcast and multicast refer to the transmission of the information data from a single source node to many destinations. Broadcast implementation sends the same information data to the other destinations in a network, whereas a multicast system transmits the same information data to a given set of destinations, that is, not all destinations. Therefore, a unicast system might be too expensive in scenarios when massive end-user destinations exist in the network since each end-user node demands a separate path and a source node, whereas broadcast and multicast modes should be more economical to meet demand. The goal of the multicast routing problem is to seek a tree spanning a source node and destinations.

In daily life, the current communication network offers perfect information transmission and some real-time multimedia communication applications for satisfying their quality of service (QoS) requirements. The rapid progress of computer communication and multimedia network technology has made network multimedia applications such as online games, video conferencing, and distance

learning into common Internet activities. The key issues of these multimedia applications need to develop efficient and fast multicast routing algorithms. Under severe QoS constraints, multicasting technology provides convenience for these services, and simultaneously makes full use of resource utilization [1,2]. Providing a high demand of better QoS is essential for many real-time applications. The guaranteed QoS in various network applications usually considers constraints such as bandwidth constraints, delay and delay jitter constraints in order to ensure smooth transmission to destinations. The establishment of the efficient QoS multicast routing problem has increasingly become one of the most critical technologies for ensuring QoS in modern integrated networks. The main objective function is to establish the optimal multicast routing tree covering the source and the multiple destinations while satisfying all QoS requirements.

The cost of a QoS multicast tree is equal to the cost sum of all the edges in this tree. A Steiner tree is needed to seek such a tree QoS multicast tree in the network. It is proved that an NP Complete can build a feasible Steiner tree [3] belongs to a NP Complete. A Steiner tree that considers QoS constraints is defined as a constrained Steiner tree. Therefore, it is also an NP complete problem [4] to construct a constrained Steiner tree. Many researchers have been interested in using different methods to solve the QoS multicast routing problem [5]. Generally, there are exact methods and heuristic algorithms to solve this problem. Although important advances have been made in the development of exact and heuristic algorithms for solving the QoS multicast tree, the exact methods to solve the multicast routing problem are a dynamic programming algorithm by Chow [6] and a branch-and-bound algorithm [7]. These two algorithms are the more practically exact methods to solve the multicast routing problem.

However, exact algorithms are still unable to solve large scale problems because of the high computational complexity, so in practice, researchers have to pay more attention to heuristic algorithms in the search for a near optimal solution. Heuristic algorithms such as meta-heuristic algorithms are actually the best choice to address the multi-constrained routing problem. They are characterized by simplicity, flexibility, and significant effectiveness. Since meta-heuristics are generally easy to implement, many researchers often adopt them to solve many complex combinatorial optimization problems. Many researchers have concentrated on adopting meta-heuristic algorithms [8] such as simulated annealing (SA) [9], genetic algorithms (GA) [10,11], bee colony algorithm [12], Cuckoo Search algorithm [13] and particle swarm optimization (PSO) [14] for solving the constrained routing problem.

The ant colony algorithm (ACO) is widely used among these meta-heuristic algorithms. Dorigo, Maniezzo and Colorni [15] proposed the first ACO algorithm, and they successfully applied it to solve the traveling salesman problem by finding the shortest path capabilities of ants. The ACO algorithm simulates the foraging behavior of ants in nature. The ants should be able to find the shortest possible route from a given food source to their nest. At the same time, these ants can adapt to changes of the surrounding environment. When they encounter new obstructions on the old shortest route path, the ants can seek the next shortest route path. This is the primary reason information is transferred among ants through pheromone trail. It is remarkable to notice that some ants by choosing the shorter route path can more quickly reconstruct the pheromone trail than those selecting the longer one. Thus, there are a greater deal of pheromone trails on the shorter paths, which can attract more ants to choose. Therefore, the Ant's foraging behavior indicates the positive feedback pheromones. The more the ants choose a specific path, the more likely that other ants choose this path later. Because of the complexity of the multicast routing problem, the basic algorithms can't meet more requirements. Some researchers have adopted different strategies to enhance basic ACO algorithm performance. Tseng et al. [16] present an ACO algorithm for solving the delay-constrained broadcasting problem, and they verify the efficiency of the algorithm by a series of experiments. Wang et al. [17] also present an ACO algorithm with consideration of the orientation factor to solve the multicast routing delay problem. Compared with the basic ACO, the solution quality and convergence rate of the improved algorithm are all enhanced. Wang et al. [18] proposed an adaptive multi-QoS routing algorithm combining ant quantity system with ACO and improving the ACO algorithm based on a routing strategy by concentrating on a dynamic topology network graph and concave metric QoS

constraint to avoid networks congestion. Wei et al. [19] adopt a hybrid updating strategy based on iteration and global optimal solutions to improve the performance of ACO algorithm. Wang and Xie (2000) [20] presented an algorithm that paid attention to the applications of the basic ACO algorithm to solve the multicast routing problem under the delay constraint. A routing table which records routings from a source node to destination nodes is built for every pair of source-destination nodes. Each ant chooses a route for one destination node. Then combine all the destination node routes and remove the overlapped edges. A tree connecting all the source and destination nodes should be built, and this tree satisfies the delay constraint simultaneously. The pheromone trail intensity on edges of the multicast tree is updated. Yin et al. [21] presented a niched ACO algorithm for solving the QoS multicast routing problem with respect to the delay and bandwidth requirements. The niched ACO algorithm first constructs a QoS constrained tree, which guarantees feasible searches with consideration of QoS requirements. The computational results show that the niched ACO algorithm outperforms the genetic algorithm on the cost. However, the algorithm only takes into account delay-and-bandwidth constraints, without considering delay jitter constraint, packet loss rate constraints, etc. Chen et al. [22] present a difference-elite ant colony algorithm, which can optimize the parameters of ACO algorithm by differential evolution algorithm so as to increase the convergence speed of the ACO algorithm. Simulation results show that the new algorithm gains minimal entropy.

Although different versions of improved ACO for the multicast routing problems have been proposed in the literature described as above, these improved algorithms mainly include three aspects. First, path selection is a main part of ant colony algorithm, so many researchers focus on designing transition rules for ants. The effects of various transition rules on ACO algorithm were investigated in [16,17,19]. Second, the pheromone updating strategy is a deterministic factor in the performance of the ACO algorithm, as shown in references [18,20]. The pheromone updating problem is how the ants update the pheromone intensity of their path. Third, the development of the ACO algorithm has led to considerable progress, but also has its strengths and weaknesses. Therefore, much research has tried to enhance the performance of hybrid algorithms [22] and new meta-heuristics, such as a bee colony algorithm [12] known as the Cuckoo Search algorithm [13]. The main weaknesses of the ACO algorithm are the slow convergence speed at the beginning of the step, and that it spends a long time on converging. While considering applying the ACO algorithm for solving the QoS multicast routing problem, the common characteristic of these algorithms is to seek the shorter paths between a source node and every destination. Then these paths are merged into a tree, and the pheromone trails on the edges of the optimal tree are updated. To improve the performance of the ACO algorithm, we present a hybrid ACO algorithm to solve the QoS multicast problem.

In this study, we seek to enhance the performance of the ACO algorithm by using the cloud model (CM) [23]. Since high pheromone trail on some edges may lead to fast convergence and fall into local optimum, we have designed the pheromone trail updating strategy based on the CM to avoid overuse of some nodes. In this paper, the main features are different from the previous ACO-based multicast routing methods described above in the following ways. First, the primary innovative point is to combine the solution generation process of the ACO algorithm with the CM, which results in a novel hybrid algorithm, namely, ACO&CM. Within the framework structure of the ACO, we embed a cloud model in the ACO algorithm to enhance the performance of the ACO algorithm by adjusting a pheromone trail on the edges. Second, although high pheromone trail intensity on some edges may trap into local optimum, the pheromone updating strategy based on the CM is used to search for high-quality areas. To avoid the generation of loops hindering a feasible multicast path construction, we adopt a memory detection search (MDS) strategy that uses two kinds of data memory structures, that is, the current list (*CL*) and the path list (*PL*). The *CL* records nodes which the current path routing reaches to avoid forming the self-loop, whereas the *PL* stores all the finished routing paths to avoid the generation of the inter-loop. Finally, computational results show that the proposed algorithm has the advantages of efficient and excellent of performance for the quality of the solution.

The paper is organized as follows. At first, the QoS problem description is given in Section 2. In Section 3, first, ACO algorithm is presented, and then the cloud model is introduced briefly. Section 4 introduces details of the proposed hybrid algorithm. Then, the results of numerical experiments are shown in Section 5. Finally, the Section 6 describes the conclusion remarks.

## 2. Problem Description

In general, a communication network may be defined as a graph $G$, $G = (V,E)$, in which $V$ represents a set of transmission nodes, $V = \{v_1, v_2, \ldots, v_n\}$, and $E$ is the set of all edges, $E = \{(i,j) \mid v_i, v_j \in V\}$ denoting physical link of these nodes. We define $n = |V|$ as the network node number, and $l = |E|$ as the network edge number. Each edge is bidirectional, i.e., the edge $e = (u, v)$ between node $u \in V$ and node $v \in V$ represents existence of another edge $e' = (v, u)$ between node $v$ and node $u$. For each edge $e \in E$, let us make some definitions about some metrics: transmission cost $C(e)$: $E \rightarrow R^+$, edge delay $D(e)$: $E \rightarrow R^+$, bandwidth $B(e)$: $E \rightarrow R^+$, transmission delay jitter $J(e)$: $E \rightarrow R^+$, packet loss rate $PL(e)$: $E \rightarrow R^+$, in which $R^+$ denotes a set of non-negative real numbers. Let $D(e)$ be the transmission delay on link $e$, and $B(e)$ denote the bandwidth functions of edge $e$. Packet loss $PL(e)$ represents the loss rate of a receiving termination on edge $e$. Delay jitter $J(e)$ denotes the delay change in different time intervals between the packets arriving. The cost $C(e)$ represents the cost of transmitting a packet on edge $e$, and it may take a measurement to make full use of resource optimization. For multicast network communications, information messages should be delivered starting from a source node to every destination node. Let a node $s \in V$ be a transmission source node. A subset $M \subseteq V$-$\{s\}$ represents destination nodes. Given $m = |M|$ is the destination node number. Let the set $M$ be a destination set, $\{s\} \cup M$ be a multicast group, and $T(s, M)$ denote a QOS multicast tree. The multicast tree represents a subgraph of weighted graph $G$ spanning the nodes in $\{s\} \cup M$. $T(s, M)$ may exist Steiner nodes. The Steiner nodes belong to a multicast tree, but these nodes do not in $\{s\} \cup M$. $p_T(s, d)$ represents a route path in a tree $T$ between a node $s$ and a destination $d \in M$, one of destination nodes.

As an example, consider the graph of Figure 1. The network graph has 9 nodes with node 0 shaded in the Figure 1a being the transmission source node, and $M = \{d_1, d_2, d_3\}$ being the destination node set. The destination nodes are marked with yellow shadows. A tree $T(s, M)$ with solid lines is the minimum spanning tree of sub-graph induced by $\{0, 2, 4, 5, d_1, d_2, d_3\}$. The multicast tree includes three steiner nodes number 2, 4, and 5, see Figure 1b.

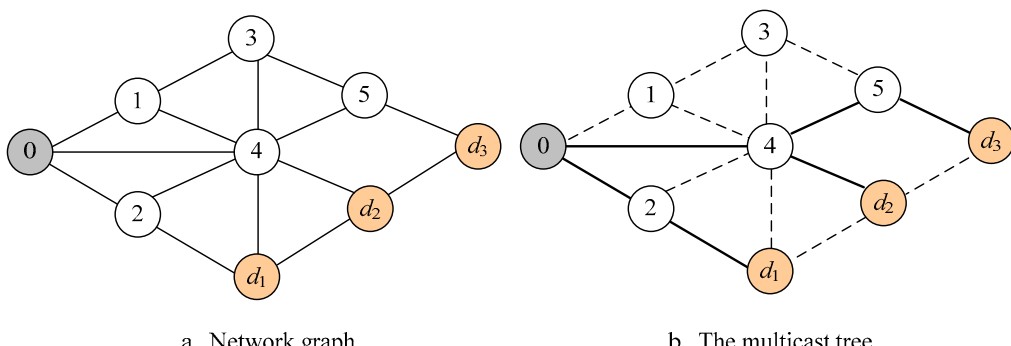

a. Network graph          b. The multicast tree

**Figure 1.** Example of constructing multicast tree.

$T(s, M)$ represents a QOS multicast tree. Let the cost of $T(s, M)$ be the cost sum of its edges, and it be described as:

$$C(T(s, M)) = \sum_{e \in T(s,M)} C(e) \tag{1}$$

In this paper, we aim to minimize costs. Thus, the objective function of this paper is to seek a tree $T(s, M)$ to minimize costs of $C(T(s, M))$ with the QoS constraint condition being guaranteed. For all the above descriptions, the model of QoS multicast problem should be expressed as the following:

$$\min C(T(s, M)) \tag{2}$$

$$\text{s.t.} \sum_{e \in p_T(s,d)} D(e) \le D_d \tag{3}$$

$$\min\{B(e), e \in p_T(s,d)\} \ge B_d \tag{4}$$

$$\sum_{e \in p(s,d)} J(e) \le J_d \tag{5}$$

$$1 - \prod_{e \in p_T(s,d)} (1 - PL(e)) \le PL_d \tag{6}$$

In this formulation, delay $D_d$ is an upper limit between a source node and a destination. Constraint (3) guarantees that the delay total of all edges along $p_T(s, d)$ should be less than the upper bound limits. Let $B_d$ be the bandwidth requirement. To guarantee the existence of feasible paths, the minimum residual bandwidth on any edge along $p_T(s, d)$ must satisfy Constraint (4). Let $J_d$ (jitter) be the packet delay variation, and $PL_d$ the packet loss rate. Constraint (5) ensures that the jitter sum on $p_T(s, d)$ cannot exceed the predefined upper limits. Constraint (6) is the constraint of packet loss. The goal of this study is to minimize costs of $T(s, M)$ in total. As we shall see, the ACO is flexible enough to handle more complicated objectives.

## 3. Ant Colony Optimization and Cloud Model

The ACO algorithm [24] is a popular-based search technique that simulates the foraging behavior of ants in nature as they find food. When the ants are foraging for food, they deposit a pheromone trail on the paths they have already passed. With the aid of these pheromones, these ants communicate and work together with others to discover the shortest route between their nests and food locations. As the ants move, the constant pheromone trail they deposit can attract other ants to follow them. The faster the pheromone trail increases, the more likely that other ants will choose that route. Over time, more ants can complete the shorter path on which pheromone trail accumulates faster. In contrast, pheromone trails on the longer routes have less pheromones deposited. In the end, the shortest route will be discovered. Ants can always find efficiently the shortest path from food sources to nests. Also, they can change the path to adapt to different environments. When new obstacles appear, they can seek the new shortest route.

The cloud model [25] is a new transition model that can be considered for both qualitative and quantitative techniques according to both fuzzy and probability theory. It adopts a new method to illustrate the fuzziness and randomness of concepts. Because the model can effectively integrate fuzziness and randomness, we can use numerical characteristics for instance, expectation value *Ex*, entropy *En*, hyper entropy *He* to generate cloud drop with certainty degree by devising ACO algorithms. Suppose $U$ is defined as the discourse universe, and $C$ denotes a qualitative concept expressed by the characters (*Ex*, *En*, *He*) related with $U$. $\mu(x) \in [0, 1]$ denotes a stable tendency for any number $x \in U$. If a number $x$ represents a random instance of the concept $C$ and accords with normal distribution $x \sim N(Ex, y_i)$, where $y_i$ also consistent with normal distribution $y_i \sim N(En, He)$, and the determinate degree of $x$ meets Equation (7). So $x$ is named as a cloud drop. If the $\mu(x)$ distribution is normal in $U$, it is named as the normal cloud.

$$u(x) = e^{-\frac{(x - E_x)^2}{2y_i^2}} \tag{7}$$

To illustrate the transformation, the model of normal cloud contains three numerical features, that is, expectation value *Ex*, entropy *En*, and super entropy *He*. The *Ex* denotes the expectation of cloud drops which can be the most representative concept belonging to the qualitative concept. Considering for a center of cloud gravity, entropy *En* means the indeterminate measurement of the qualitative concept according to the arbitrary probability and fuzziness. Super entropy *He* denotes the dispersed degree of the entropy *En*, representing the dispersion extent of cloud droplets. Figure 2 shows the diagram of the cloud model. It can be seen that the greater *En* represents the greater coverage of the cloud drops. Similarly, the greater the *He*, the more dispersed the cloud drops are.

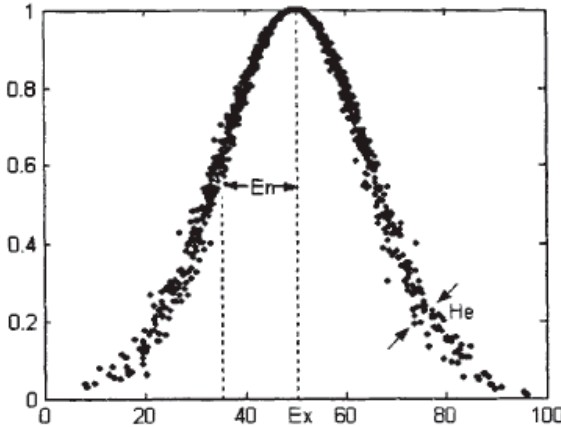

**Figure 2.** Diagram of the Cloud.

## 4. ACO&CM Algorithm

In order to overcome the drawback that the ACO algorithm may easily fall into local optimum, we designed our solution methodology to solve the QoS multicast routing problem. Since solving this problem is rather complicated, this can lead the solution method into more difficult conditions. High quality solutions should require us to design the algorithm elaborately. To enhance the performance of the ACO algorithm, we present the hybrid ACO algorithm, namely, the ACO&CM algorithm, which can effectively integrate the solution building mechanism of the ACO algorithm and CM. Meanwhile, the cloud model embedded in the ACO framework can avoid overuse of some nodes and the algorithm easily getting trapped in a local optimum by adjusting the pheromone trail. This method has the outstanding advantage of the ACO algorithm, which can find better performance solutions. It also has the advantage of CM, which is capable of searching different solution spaces in order to get better solutions. To deal with the probability of forming a loop in the search path, we devise a memory detection search (MDS) strategy, and incorporate the MDS strategy into the path construction process. Furthermore, in order to obtain high quality solutions, we adopt improvement strategies by deleting an edge with the maximum cost to create a neighboring solution of the original tree. The framework of proposed algorithm contains a memory detection search strategy, path construction, tree construction, pheromone trail updating, and solution improvement.

### 4.1. Memory Detection Search Strategy

Many literatures provide heuristic algorithms for constrained tree construction. These heuristic algorithms mainly include two types. The first type is to build a spanning tree with minimum-cost. Then, to make it satisfy the corresponding requirements, the tree structure should be pruned. The second type is to separately generate the constrained routing path starting from a source to every destination. After that, one main path is determined as the main fame tree, and other paths should be integrated into the tree. When the paths are constructed from a source node to destinations, there are two types of forward and backward path construction. The forward path construction starts the path search from a source and ends at the destination. The backward path construction proceeds

from destinations to a source. To avoid forming loops, we adopt a memory detection search (MDS) strategy, and embed it into the process of tree construction. The MDS strategy adopts the second category to construct trees and we use the forward path construction.

The MDS strategy can ensure the search towards any feasible path considering QoS constraints. To avoid the generation of loops influencing on feasible multicast path construction, we adopt a memory detection search (MDS) strategy that uses two kinds of data memory structures: the current list (*CL*) and the path list (*PL*). The *CL* records nodes, which the current path routing reaches during the depth-first search process to impede the self-loop generation, while the *PL* stores all the built routing paths to avoid the generation of the inter-loop.

Figure 3 illustrates the process for avoiding generating the self-loop. The network graph has nine nodes, in which node 0 is a source node, and $M = \{d_1, d_2, d_3\}$ represents the destinations. At the beginning of the path construction, the search process starts from source node 0 and selects the next nodes based on the proposed ACO algorithm mechanism. When there are no given destinations to choose from, the search process will not continue going with a deeper search without returning nodes in *CL*, and the search progress should be traced back to one level before starting another search branch. Meanwhile, the MDS strategy records visited nodes in *CL* during the current path construction. In Figure 3a, assume the path process arrives at node 1. Then, all the nodes {0, 4, 5, 3, 1} on the current path have been recorded in *CL*. Because node 4 and node 0 have been recorded in *CL*, the two nodes are not accessible to avoid the self-cross loop. There are no other nodes from the current node 1, so the traceback operation is executed by considering nodes in *CL* till the routing path process returns to node 5 as given in Figure 3b,c. When tracing back to node 5, the path process searches for destination node $d_3$, completes the path search, and records the path in *PL*, i.e., $PL = \{\{0, 4, 5, d_3\}\}$ as illustrated in Figure 3d.

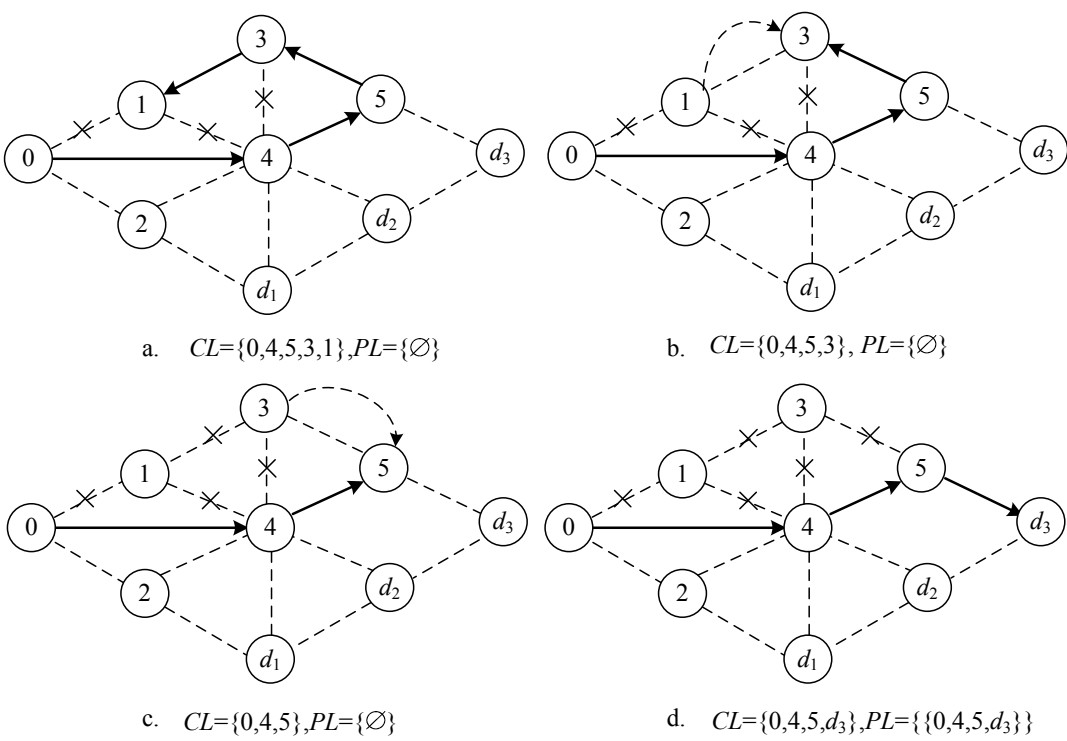

a.   $CL=\{0,4,5,3,1\}, PL=\{\varnothing\}$            b.   $CL=\{0,4,5,3\}, PL=\{\varnothing\}$

c.   $CL=\{0,4,5\}, PL=\{\varnothing\}$            d.   $CL=\{0,4,5,d_3\}, PL=\{\{0,4,5,d_3\}\}$

**Figure 3.** An instance of producing the self-loop.

We adopt the MDS strategy for avoiding the generation of the inter-loop using *PL*. Figure 4a shows an example, in which a path {0, 4, 5, $d_3$} has been built. The *CL* records visited nodes in the process of the current path construction. Before starting a next path, store *CL* to *PL* and set $CL = \varnothing$. The *PL* stores the nodes constituting all previously built paths, $PL = \{\{0, 4, 5, d_3\}\}$. We can begin with

the next path. Suppose in Figure 4a that the routing process begins the path search from a source and moves to node 2. In the scenario, because node 4 is recorded in *PL,* it is not accessible to avoid forming the inter-loop, which is formed by the current path and the completed paths. Next, the path construction process points to node $d_1$ directly, one of the destinations, which has not added to the multicast route paths as shown in Figure 4c, and then a current routing path *CL* = {0, 2, $d_1$} has been built. Add *CL* to *PL,* that is *PL* = {{0, 4, 5, $d_3$}, {0, 2, $d_1$}}. Then, we can proceed with the routing for the next path starting from node 0 and proceeding to the next destination. At this time, we have no way to extend the current path without considering the nodes in *PL* that have been visited. When the routing path process aids a constructed path, we have to trace the nodes that are in *PL* can be accessed again. Although node 4 is contained in *PL,* there are no other nodes to choose from the source node 0. The routing path process would have to proceed back to node 4. Then, let the current routing path process directly move to $d_2$, the last node of the destinations, and the current path is completed as {0, 4, $d_2$} (see Figure 3d). Add *CL* to *PL,* that is *PL* = {{0, 4, 5, $d_3$}, {0, 2, $d_1$}, {0, 4, $d_2$}}. When all the destinations have been added to the multicast route paths, the path search stops.

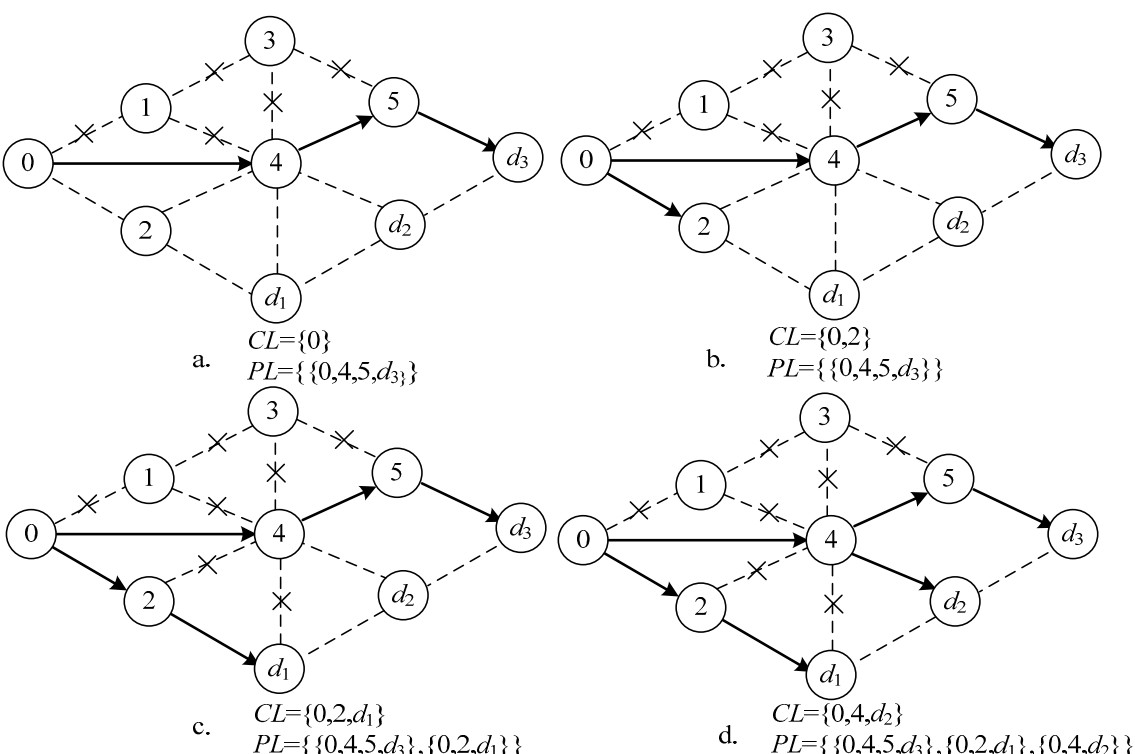

**Figure 4.** An instance of producing the inter-loop.

*4.2. Path Construction*

In the process of constructing paths, ants sequentially find the minimum cost path proceeding from a source node *s* to destination nodes. It is necessary for one path to determine a nice sequence of nodes with the least cost. Each individual ant starting at the source node *s* simulates a path. The ants sequentially choose nodes to construct paths. Initially, the ant *k* successively chooses the node *j* and forms the sequence of nodes in the path until the ant completes the path. The selected nodes should first satisfy QoS multicast route constraints in the delay, residual bandwidth and the delay jitter as defined in constraints (3), (4) and (5), and then packet loss as described in constraints (6). If a node violates one or more of the constraints, the node should not be chosen, and the next node will be

selected to check whether the constraint condition is satisfied. When ant $k$ is building the path, at the node $i$ the ant would select node $j$ according to the following rules:

$$
j = \begin{cases} \underset{u \notin M_k}{\mathrm{argmax}} \left\{ \tau_{il} \cdot [\eta_{il}]^\beta \right\}, \text{ if } q \leq q_0 \\ S, \text{ otherwise} \end{cases}
\tag{8}
$$

in which $\tau_{il}$ represents pheromone trail concentration between $i$ and $j$ nodes, and $\beta$ represents a parameter determining the comparative effect of visibility. The value $q$ denotes a random variable, and it is a uniform value in [0, 1]. $q_0 \in [0, 1]$ is a user-specified parameter value. The visibility $\eta_{il}$ is an inverse of an edge. $M_k$ is the collection set including all the nodes that the ant $k$ has already visited. Other ants cannot select the nodes in the set $M_k$. If the $q \leq q_0$ condition is satisfied, an edge with the minimum cost is determined according to Equation (8); otherwise, choose an edge depending on $S$. $S$ is a random variable using the random distribution given as the following:

$$
P_{ij}^k = \begin{cases} \dfrac{[\tau_{ij}]^\alpha [\eta_{ij}]^\beta}{\sum\limits_{l \notin M_k} [\tau_{il}]^\alpha [\eta_{il}]^\beta}, j \notin M_k \\ 0, \text{ otherwise} \end{cases}
\tag{9}
$$

where $\alpha$ is a parameter. $\alpha$ denotes the relative effect of the pheromone concentration, indicating the influence of accumulated pheromone trail in the process of ant movement. The bigger the value of $\alpha$, the more inclined other ants are to choose the same paths. $p_{ij}^k$ is the transition probability of the ant $k$. In ACO algorithms, the path construction process is similar to a greedy rule, except for choosing the next node by the probabilistic formula rule rather than the deterministic one.

In the process of constructing the node sequence of the path, each individual ant not only considers the QoS constraint conditions, but also adopts the MDS strategy to avoid the generation of a loop. The MDS strategy uses the current list (*CL*), and the path list (*PL*) memory structures to avoid the generation of a loop. The *CL* records current path nodes in case of forming the self-loop, and the *PL* stores all the routing paths to avoid the generation of the inter-loop. Every path is put in *PL* set and every element represents a path in this set. $N_{ant}$ denotes the number of ants, which is the same with a destination node number. Every ant simulates a path starting from a given source node to a destination. In the process of constructing the paths, these paths are searched by the ACO mechanism and every path should satisfy the QoS constraint. Path construction procedure steps are given as the following:

**Step 1**: Initialize $CL = \varnothing$; path set $PL = \varnothing$; Initialize set $S = \{s\}$; $S_m = M$, $N_{ant} = |M|$;

**Step 2**: Select randomly an ant from the ant set $N_{ant}$;

**Step 3**: Set $CL = \varnothing$;

**Step 4**: Let the source node $s$ be the current point, $v_i = s$. Each ant begins from the node $v_i$ to find the path to the destination node;

**Step 5**: Each ant chooses the next node $v_j$ which connects with current node using formula (8). Every path satisfies the QoS constraint conditions.

**Step 6**: Check self-loop or inter-loop. If $v_j$ is neither in set *CL* nor in set *PL*, then add $v_j$ to set $CL = CL \cup \{v_j\}$; otherwise go to Step 5;

**Step 7**: If $v_j$ is included in $S_m$ then add $v_j$ to set $CL = CL \cup \{v_j\}$, and move $v_j$ from $S_m$, $S_m = S_m / \{v_j\}$, add the *CL* to *PL*, $PL = PL \cup CL$; otherwise, the node $v_j$ is regarded as the current node, $v_i = v_j$, go to Step 5;

**Step 8**: If the termination condition $S_m = \varnothing$ is met, all the destination nodes are put into the paths and $N_{ant}$ paths have been constructed; otherwise, go to go to Step 2.

### 4.3. Tree Construction

Though the paths discovered cover a source node and destination nodes, these paths can not directly form a multicast tree because they include some repeated edges. Therefore, to obtain a real multicast tree, we should prune the paths to remove these repeated edges. The completed routing paths *PL* is defined as the input parameter of tree construction. Let $V_T$ be a node set, and *T* be an edge set. Randomly choose a route path from the path set *PL* and add the nodes and edges of this path to $V_T$ and *T*, respectively. Repeat this procedure until the terminating criterion is met. The output parameter would be a multicast tree spanning a source nodes *s* and destinations in *M*. The steps of tree construction procedure should be described as the following:

**Step 1**: Initialize $T = \varnothing$, $V_T = \varnothing$;

**Step 2**: Choose a route path *p* randomly from the set *PL* and suppose *p* is defined as $v_0, v_1, \dots v_k$ node sequence;

**Step 3**: $i = 0$;

**Step 4**: Check whether the node $v_i$ is in $V_T$. If the node $v_i$ is in $V_T$, then go to Step 5; otherwise, put $v_i$ into $V_T$, $V_T = V_T \cup \{v_i\}$, and add the edge $(v_i, v_{i-1})$ to *T*, $T = T \cup \{ (v_i, v_{i-1}) \}$;

**Step 5**: $i$++;

**Step 6**: Check whether the condition $i < k$ is met, if $i < k$, then go to step 4;

**Step 7**: Delete *p* from *PL*, $PL = PL/\{p\}$;

**Step 8**: If the termination condition $PL = \varnothing$ is met, and then return to *T*; if not go to Step 2.

### 4.4. Pheromone Trail Updating

In ACO, there are local and global updates in pheromone updating. Local pheromone trail updating is executed while constructing a multicast tree, and global pheromone updating is executed at the end of completing the multicast tree. The main purpose of local pheromone updating is to avoid producing very high pheromone links being selected by other ants. These links use more pheromone on the trail to make the ACO algorithm fall into a local optimum. The local pheromone trail is updated using the following formulation:

$$\tau_{ij} = (1 - \rho)\tau_{ij} + \Delta\tau \tag{10}$$

where $\rho \in [0, 1]$ represents a pheromone evaporation parameter, $\Delta\tau = \sum\limits_{k=1}^{m} \Delta\tau_{ij}^k$, $\Delta\tau_{ij}^k$ is the incremental pheromone trail on the edge between *i* and *j* nodes, and *m* means the ant number. To avoid overuse of some nodes and hinder the ants from exploring new paths, we adjust parameter $\rho$ by cloud model to avoid the ACO algorithm trapping into local optimum. The parameter $\rho$ is performed using the following formula,

$$\rho = \begin{cases} k_1 e^{\frac{-(x - E_x)^2}{2y_i^2}}, & \tau_{\min} < x < \tau_{\max} \\ \rho_1, & x > \tau_{\max} \\ \rho_2, & x < \tau_{\min} \end{cases} \tag{11}$$

where $\rho_1, \rho_2 \in [0, 1]$ are pheromone decay parameters, and $k_1$ denotes a parameter. Other parameters refer to the description of formula (7).

Besides, the pheromone of global solution is updated according to global pheromone updating, which is intended to strengthen the neighborhood search of the optimum solution. The global pheromone updating can not only accelerate convergence by increasing differences between better and worse solutions, but also avoid the fast accumulation of pheromones on relatively optimal paths at an early stage. In the basic ACO algorithm, only the pheromone trails on the edges of the best solution have the chances to be updated using the following formula,

$$\tau_{ij} = (1 - \varphi) \cdot \tau_{ij} + \frac{\varphi}{L_{best}} \tag{12}$$

in which $\varphi \in [0, 1]$ is a decay parameter, $L_{best}$ is the cost value of the best-so-far solution.

### 4.5. Solution Improvement

After the feasible solution has been obtained, we try to enhance the solution quality by using improvement strategies. Figure 5 illustrates the process of solution improvement. Figure 5a illustrates an example where a multicast tree has been built. The primary concept behind improvement strategies is trying to seek improved solutions by deleting an edge with the maximum cost. Then a multicast tree is split into two parts of $\{0, 2, 1, 4, d_2, 3, 5, d_3\}$ and $\{d_1\}$ by deleting edge $(2, d_1)$ with the maximum cost as shown in Figure 5b. Choose a node from the part, in which there is no source node. Extend the node to link the part in which the source node exists until a new tree is produced, see Figure 5c. The new tree must satisfy the multicast constraint. Although the new tree spans all the source node and destination nodes, it does not belong to a multicast tree because there are some excrescent edges after connecting the tree, and the nodes connected by excrescent edges have some leaf nodes which are not multicast members. Therefore, to obtain a true multicast tree, we should prune the leaf nodes, which do not belong to the multicast members in Figure 5d.

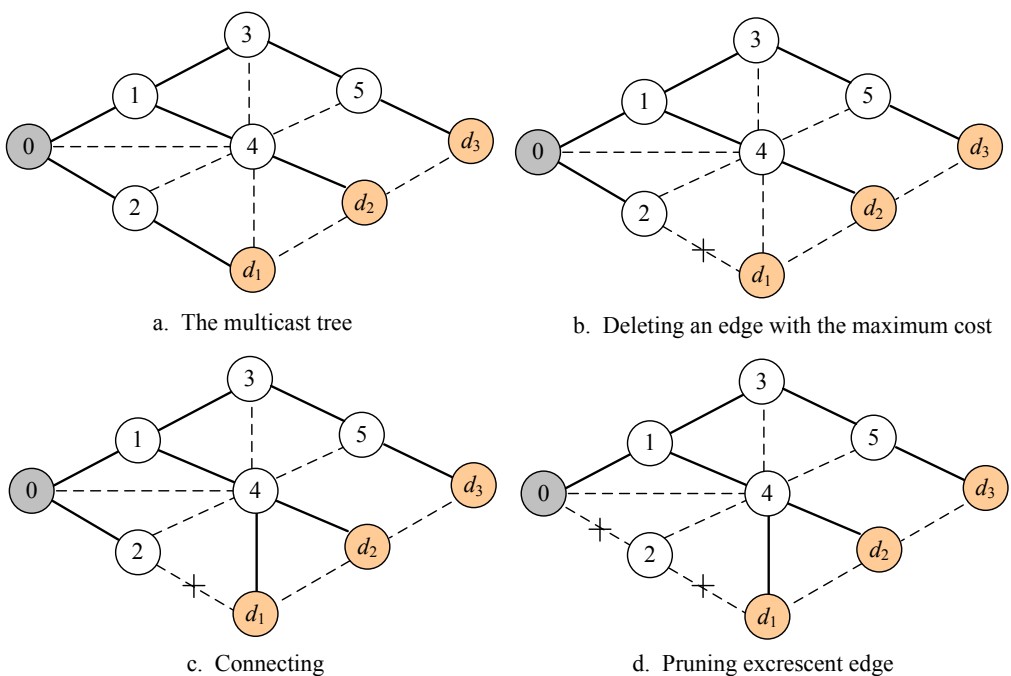

a. The multicast tree

b. Deleting an edge with the maximum cost

c. Connecting

d. Pruning excrescent edge

**Figure 5.** Illustration of solution improvement process.

The framework of the ACO&CM algorithm consists of two main procedures which connect the parts together, including as the MDS strategy, pheromone updating, and solution improvement. In this study, the two main procedures are called path construction and tree construction, respectively, which are developed to build feasible multicast trees. The first procedure basically starts from a source node to search for the paths of destinations and each path should satisfy the Qos constraint. Then, the second procedure integrates *m* routing paths obtained by applying the first one to build a multicast tree *T*. The stopping criterion consists of the convergence rule and the maximum iterations. The process stops when the stopping criterion is satisfied. The main steps of hybrid algorithm would be formulated as below:

**Step 1**: Initialization. Input the pheromone trails on all edges, edge cost *c(e)*, link delay *d(e)*, link bandwidth *b(e)*, and $D_d$, $B_d$, $J_d$, and $PL_d$ parameters. Let *s* be the starting node, and $M = \{d_1, d_2, \ldots, d_m\}$ as destination nodes. Set population number as *TreeNum*. Set *iter* = 0 where *iter* counts the number of iteration and will be compared to the maximum number of iterations, $CN_{\max}$.

**Step 2**: Initialize the tree set $S_T = \varnothing$;

**Step 3**: Execute path construction procedure to obtain $m$ routing paths starting from the node $s$ to destination nodes. Record the completed routing path in *PL*.

**Step 4**: Execute tree construction procedure to produce a multicast tree $T$, which is created by merging $m$ paths. Store the multicast tree in $T$. Update local pheromone trail accord to formula (10).

**Step 5**: Improvement solutions.

**Step 6**: Calculate the fitness values of $T$ and check whether to update the current best solution $T_{bestsol}$. If condition $C(T) < C(T_{bestsol})$ then $T_{bestsol} = T$.

**Step 7**: Add $T$ to $S_T$, i.e., $S_T = S_T \cup T$.

**Step 8**: Repeat Steps 3–7 until $|S_T| > TreeNum$.

**Step 9**: Update global pheromone trail on the edges of $T_{bestsol}$ using formula (12).

**Step 10**: Set *iter* = *iter* + 1;

**Step 11**: Repeat Steps 2–11 until the algorithm converges or *iter* > $CN_{max}$.

## 5. Computational Results

We present an experimental study to verify the performance of the hybrid algorithm. We have implemented a basic ACO algorithm and hybrid algorithm using visual C++. To evaluate the effectiveness of the ACO&CM algorithm, we have carried out numerical experiments on the network topology for the QoS routing problem. Figure 6 shows a test network topology with 30 nodes. Some parameter values in the ACO&CM algorithm directly or indirectly influence the final results. Therefore, these parameter values may be measured on the test instance. We set parameter values as follows: $q_0 \in [0.65, 0.80]$, $\beta \in [3, 4, 5]$, $\alpha = \rho \in [0.35, 0.50]$, *TreeNum* = 30. Notably, for each edge, the initial value of $\tau_{ij}$ is a very small constant, $\tau_{ij} = (n \cdot L_{best})^{-1}$, where $n$ denotes the node number of multicast tree, and $L_{best}$ represents the minimum cost of QoS multicast tree, and. All the experiments should be executed for $CN_{max} = 1000$. The edges are generated randomly. $c(e)$ is defined as actual distance between two nodes of the edge $e$. $d(e)$ and $b(e)$ of the edge $e$ are set to sqrt($c(e)$), $c(e)/2$, respectively.

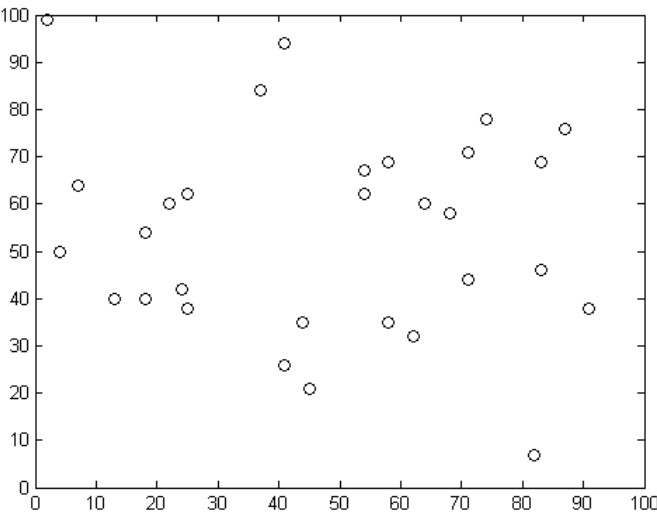

**Figure 6.** A test network topology.

To investigate the convergence properties of ACO and ACO&CM algorithms, we test their convergence performance on the 30 nodes. Due to publishing space restrictions, we only provide the convergence performance of two algorithms averaged over 20 runs on this test problem. Figure 7 traces the dynamic changes of best cost with the iteration variation for two algorithms. From the results in Figure 7, the convergence process of ACO&CM is shown to be faster than that of ACO. It demonstrates that ACO&CM algorithm has a higher search ability than the ACO algorithm on this problem.

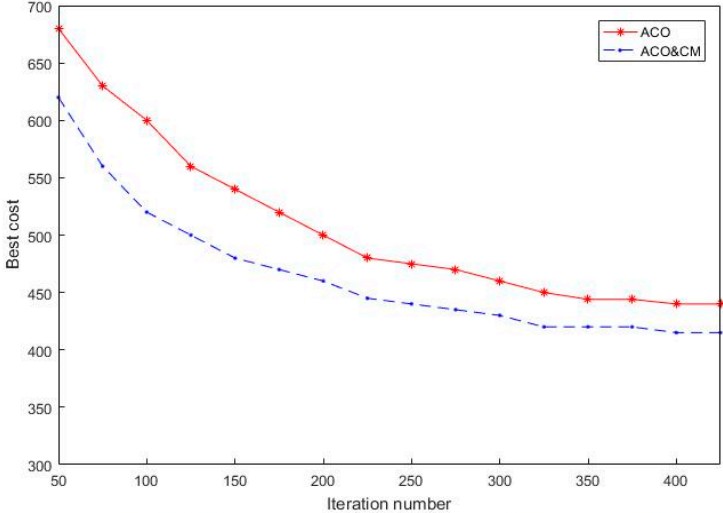

**Figure 7.** Comparison of the convergence of ACO and ACO&CM.

To evaluate the stability of our proposed ACO&CM algorithm, we further test the algorithm independently 20 runs on the problem instances with 30 nodes. For each independent run, the best cost values of the objective function are depicted in Figure 8. From the result in Figure 8, it can also be observed that ACO&CM yields a much lower cost than that produced by ACO. Both of ACO&CM and ACO algorithms can obtain a feasible solution in 20 runs. Once again, this result illustrates the effectiveness of ACO&CM.

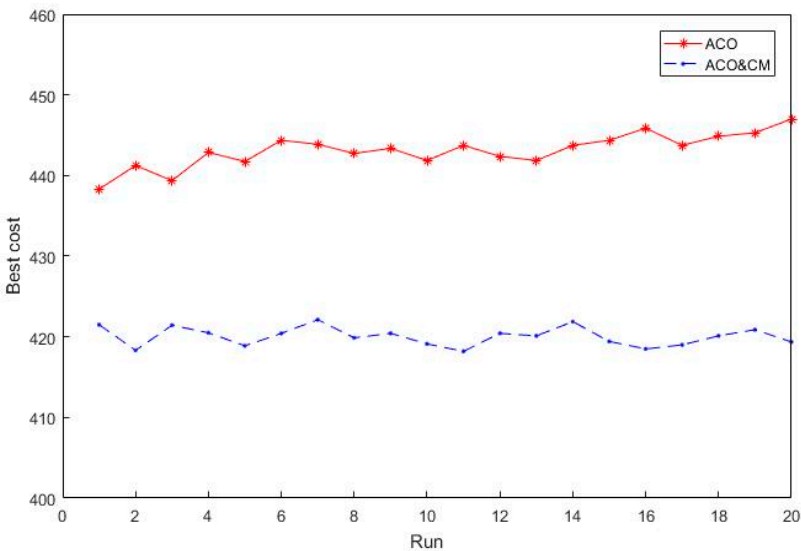

**Figure 8.** Comparison of the cost of ACO and ACO&CM.

To give further evidence concerning the conclusion, we evaluated scalability of algorithms with the instance size according to the execution time in seconds and the corresponding best cost of a multicast tree. Table 1 shows the best cost and corresponding run time of ACO and ACO&CM algorithms for the problem instance. The node number column shows the number of testing instance nodes. These testing instances are from small size with 30 nodes to large size with 175 nodes. In Table 1, ACO represents the basic ant colony optimization without combining with some other algorithms, and ACO&CM refers to the proposed algorithm. Table 1 has also shown the gap between the ACO and ACO&CM algorithms. The gap is specified as percentage improvement in the cost values of the two algorithms. Form Table 1, we discover that the hybrid proposed algorithm can yield best performance of the minimum cost objective. Concerning solution quality, we can conclude that the

average cost of multicast tree obtained by ACO&CM is much less than that by basic ACO, and the ACO&CM algorithm can easily find the better solution with significantly shorter computation times. The average improvement gap is 2.78%, and the percentage shows that our proposed ACO&CM algorithm is much better and more efficient than ACO algorithm for the minimum cost objective to solve the QoS multicast routing problem.

In order to a make further comparative analysis of the proposed algorithm's performance on the QoS routing problem, the proposed algorithm is also compared with performance of GA. The GA proposed by Wang et al. [11] adopted the coding method with a tree structure. A chromosome represented a feasible multicast tree. The GA has the following parameter values: crossover probability $p_c \in [0.8, 1]$, mutation probability $p_m \in [0.1, 0.5]$, the maximum number of iterations $CN_{max} = 1000$, population number *TreeNum* = 30. A randomized algorithm based on depth-first search was used for building a random Steiner tree to construct the initial population. Then, the multicast tree changed through adopting crossover and mutation. Since ACO and GA methods are both random algorithms, the same procedure in the same instance may generate different results due to different random numbers in each run. We thus repetitively executed each instance 20 times and the average values over these times are depicted in Figure 9. Figure 9 shows the illustration of comparison of the cost attained by ACO&CM, ACO, and GA against different numbers of nodes. It also can be shown from the curves that in nearly all scales of topology ACO&CM outperforms ACO and GA in an extremely important way. Besides, ACO yields less total cost than GA, so ACO shows a significantly better performance than GA. The computational results also prove that ACO&CM outperforms ACO, and the multicast trees obtained are feasible if these results meet the QoS constraints.

**Table 1.** The comparison of ACO with ACO&CM Algorithm.

| Number | Node Number | ACO | | ACO&CM | | % Gap |
|---|---|---|---|---|---|---|
| | | Best Cost | Time (s) | Best Cost | Time (s) | |
| 1 | 30 | 443.80 | 8.86 | 420.14 | 8.29 | 5.63 |
| 2 | 50 | 435.79 | 59.78 | 428.47 | 52.25 | 1.63 |
| 3 | 75 | 554.75 | 139.83 | 551.74 | 105.00 | 0.55 |
| 4 | 100 | 766.25 | 388.24 | 756.18 | 361.38 | 1.31 |
| 5 | 125 | 830.22 | 465.31 | 798.12 | 436.50 | 4.02 |
| 7 | 150 | 926.15 | 570.73 | 900.31 | 538.28 | 2.87 |
| 8 | 175 | 998.14 | 716.26 | 965.65 | 665.17 | 3.36 |

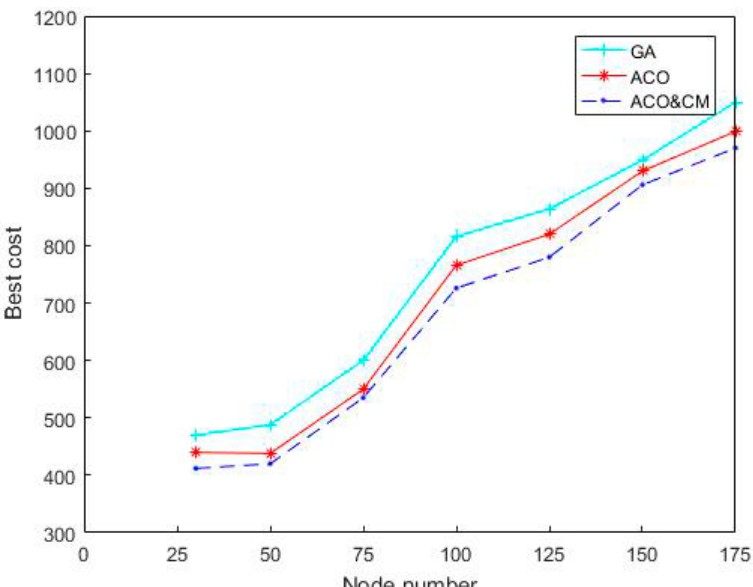

**Figure 9.** Comparison of the cost in different nodes.

## 6. Conclusions

In this article, we present a hybrid ant colony optimization (ACO&CM) for solving the QoS multicast routing problem. The primary innovation is to combine the solution generation process of the ACO algorithm with the CM. Since a high pheromone trail on some edges may lead to fast convergence and fall into local optimum, within the framework structure of the ACO, we embed a cloud model in the ACO algorithm to enhance the performance of the ACO algorithm by adjusting the pheromone concentration on edges. To avoid the possibility of loop formation in the process of constructing a feasible multicast path, we devise a memory detection search (MDS) strategy, that is, the current list (*CL*) and the path list (*PL*), and integrate it into the path construction. The simulation results have demonstrated that our ACO&CM algorithm was able to attain better or competitive performance. The main limitations in this study mainly lie in that the test instances are not large enough, and the self-adaptive adjustment of principal parameters is not considered. These key issues are to be solved in future research work. Therefore, in the future, we are planning to further improve the performance of the ACO&CM algorithm, especially in large-scale instances. We will study the comparison between this algorithm and other algorithms, such as cuckoo search, particle swarm optimization, etc. Besides, according to the search characteristics of ACO&CM, our future work includes designing a new self-adaptive adjustment strategy of principal parameters to improve the adaptability of the algorithm to different practical conditions.

**Author Contributions:** X.Z. designed the algorithms and wrote the paper; X.S. performed the experiments; Z.Y. was responsible for the reviews and discussed the results.

**Funding:** This work is supported by National Science Foundation of Liaoning (Grant No. 20170540471) and Foundation of Liaoning Educational Committee (Grant No. L2015265).

**Conflicts of Interest:** The authors declare no conflicts of interest.

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
