# Peer review of "A Novel Hybrid Ant Colony Optimization for a Multicast Routing Problem"

_algorithms, doi:10.3390/a12010018_

Round 1

Reviewer 1 Report

The motivation of the manuscript entitled: “A Novel Hybrid Ant Colony Optimization for

 Multicast Routing Problem is quite clear and rational.

This manuscript tackles a very important problem and fitting well with the journal’s scope.

The manuscript has included interesting ideas and concepts. The approach followed looks useful and the results are promising.

However, the following issues should be addressed before considering the manuscript for publication.

·       Literature review looks like more a list of references. In order to be valid, this paper MUST include a proper analysis of the relevant literature on field and then make a comparison with the authors' approach.

·       Literature review has to be more carefully done, including papers published very recently.

·       What are the main limitations of this approach?

·       The conclusions are slight. The conclusions should better explain the results obtained and the relevance of the approach followed. Furthermore, some brief indication of the thoughts of the authors on the further work would be a useful addition to conclusion.

·       As usual a final thorough proof-reading is recommended.

I would however encourage the author to pursue his efforts in improving the paper for future publication since the topic of the research is highly relevant.

Author Response

Ref. No.: 1

Response to the Referee’s Comments on Paper Algorithms-413453:

We thank the referee very much for the comments and suggestions.  They are very helpful for us to revise and improve the paper.  The paper has been carefully revised according to the referee’s advice.  We have made the following changes on the paper accordingly.

1.      We have made considerable efforts on revising the literature review.  We have given a proper analysis of the relevant literature on our study field.  We have added and discussed some recent references relevant for this topic in the following literatures.  

[1] Wei, Y.; Zhao, K. X.; Zhang, S. Q.; Wang, D. S. Research of Improved Ant Colony Algorithm in QoS Multicast Routing. Bull. Sci. Technol. 2017, 33,183–186.

[2] Chen, S.; Xu, B.; Xu, B. G. Qos multicast routing optimization algorithm based on difference-elite ant colony. Comput.Eng.2015, 41,117–125.

2.      The main limitations in this study mainly lie in that the test instances are not large enough, and the self-adaptive adjustment of principal parameters is not considered.  These key issues are to be solved in the future research work.

3.       We have revised the description of Section 6.  Meanwhile, we have added some brief indication of our thoughts on the further work.

4.       According to the referee’s advice, we have made a very careful proof-reading.

Reviewer 2 Report

The algorithm is well presented, and the results seem feasible.

The English language is generally fine. Some of the problems I’ve noticed:

Lines 61-62: Generally, there are exist exact methods and heuristic algorithms to solve this problem.

Lines: 84-86: It is remarkable to notice that some ants by choosing the shorter route path will be more quickly reconstruct the pheromone trail than those selecting the longer one.

Generally, there exist exact methods and heuristic algorithms to

In this study, we intent make great effect to enhance the performance of ACO algorithm by using the cloud model (CM) [21]. I couldn’t find the full text of cited paper [21], but from the title and from the abstract, it doesn’t seem to talk about the CM. Furthermore, the bibliography seems dated. You should discuss some newer papers relevant for this topic.

Line 169: Our research aims to minimize the cost in this paper. -> In this paper we aim to minimize the cost. The next sentence should also be rephrased.

Lines 186-187 probably sound better like this: The faster the pheromone trail increases, more probably other ants will choose the route.

Author Response

Ref. No.: 2

Response to the Referee’s Comments on Paper Algorithms-413453:

We thank the referee very much for the comments and suggestions.  They are very helpful for us to revise and improve the paper.  The paper has been carefully revised according to the referee’s advice.  We have made the following changes on the paper accordingly.

1.        We have revised the sentence.  Lines 61-62: Generally, there are exact methods and heuristic algorithms to solve this problem according to the referee’s suggestions.

2.        According to the referee’s suggestions, we have revised the sentence.  Lines: 84-86: It is remarkable to notice that some ants by choosing the shorter route path will be more quickly reconstruct the pheromone trail than those selecting the longer one.

3.        According to the referee’s suggestions, we have replaced Ref.[21] with a newer one.  

4.        According to the referee’s suggestions, we have revised the sentence.  Line 169: In this paper we aim to minimize the cost.  

5.        According to the referee’s suggestions, we have revised the sentence. Lines 186-187: The faster the pheromone trail increases, the more probably other ants will choose the route.  

Reviewer 3 Report

This work presented a hybrid ant colony optimization (ACO&CM) for solving the QoS multicast routing problem.

The cloud model is embeded in the ACO algorithm to enhance the performance of ACO algorithm by adjusting pheromone concentration on edges.

A test network topology with 30 nodes is used as test example.

Some more detailed comments are given below. I hope that if the authors will take them into account the paper will be improved.

1. There are many versions of state-of-the-art ant colony algorithm, such as enhanced ACO, adapted ACO, modifined ACO, integrated ACO, collection path ACO, Arc based ACO, niching Pareto ACO, adaptive polymorphic ACO, cellular automata-based improved ACO, dynamic ant colony's labor division, best-path-updating information-guided ACO, Fidelity-based ACO, etc.. 

Authors should survey and discuss these state-of-the-art ACO in section 1.

2. Authors should compare the performance for convergence and soluion qiality of proposed method with existing state-of-the-art ACO in Section 5.

3. Except Ref. [5]&[12], it has almost no recent references from the last two years. Recent and state-of-the-art references should be cited on the topic.

4. Provide all parameter setting of the ACO&CM, ACO, and GA in Section 5.

Author Response

Ref. No.: 3

Response to the Referee’s Comments on Paper Algorithms-413453:

We thank the referee very much for the comments and suggestions.  They are very helpful for us to revise and improve the paper.  The paper has been carefully revised according to the referee’s advice.  We have made the following changes on the paper accordingly.

1. We have made considerable efforts on revising the literature review and discussed some references relevant for this topic.

2.      Since it is a very difficult problem to solve the QoS multicast routing problem,
it took us more than a year to finish the paper. Since ACO and GA methods
are both random algorithms, we have verified the comparison between the
proposed algorithm and GA. In the future, we will study the comparison
between this algorithm and other algorithms. We are planning to further
improve the performance of the ACO&CM algorithm, especially in large-scale
instances.

3.      It is a very difficult problem to solve the QoS multicast routing problem, and we focus on the ACO algorithm relevant, hence recent references on this problem are rather scarce. We have added and discussed some recent references relevant for this topic in the following literatures.

[1] Wei, Y.; Zhao, K. X.; Zhang, S. Q.; Wang, D. S. Research of Improved Ant Colony Algorithm in QoS Multicast Routing. Bull. Sci. Technol. 2017, 33,183–186.

[2] Chen, S.; Xu, B.; Xu, B. G. Qos multicast routing optimization algorithm based on difference-elite ant colony. Comput.Eng.2015, 41,117–125.

4.       According to the referee’s advice, we have provided the parameter setting as follows:

[1] We have added the population number parameter setting of the ACO&CM, ACO in section 5.

[2] The GA has the following parameter values: crossover probability pcÎ[0.8,1], mutation probability PmÎ[0.1,0.5], the maximum number of iterations CNmax=1000, population number TreeNum =30.

Round 2

Reviewer 1 Report

The manuscript entitled: “A Novel Hybrid Ant Colony Optimization for Multicast Routing Problem” has included interesting ideas and concepts; it is well grounded in the body of knowledge in the area. The paper is well written and organized and includes many mathematical definitions to support the approach followed. The approach followed looks useful and the results are promising.

Taking into account the comments of previous reviews, the authors have made a great effort to improve it and the main weaknesses are solved. The paper is now more consistent and is quite interesting and informative to most readers.

Thus, in my opinion, the paper is recommendable for publication.

Reviewer 3 Report

The authors have carefully addressed the previous comments of the reviewer and significantly improved the manuscript.